# Co-Creation and Implementation of a Healthy Snacks Policy in Primary Schools: Data from Sintra Grows Healthy

**DOI:** 10.3390/nu16193374

**Published:** 2024-10-04

**Authors:** Telma Nogueira, Raquel J. Ferreira, Mariana Liñan Pinto, Vitória Dias da Silva, Paulo Jorge Nogueira, Joana Sousa

**Affiliations:** 1Laboratório de Nutrição, Faculdade de Medicina, Universidade de Lisboa, Avenida Professor Egas Moniz, Edifício Reynaldo dos Santos, Piso 4, 1649-028 Lisboa, Portugal; marianapinto@medicina.ulisboa.pt (M.L.P.); vitoriasilva@medicina.ulisboa.pt (V.D.d.S.); joanamsousa@medicina.ulisboa.pt (J.S.); 2Instituto de Saúde Ambiental, Faculdade de Medicina, Universidade de Lisboa, Avenida Professor Egas Moniz, Edifício Egas Moniz, 1649-028 Lisboa, Portugal; 3Câmara Municipal de Sintra, Departamento de Educação, Juventude e Desporto, Largo Dr. Virgílio Horta, 2714-501 Sintra, Portugal; raquel.ferreira@cm-sintra.pt; 4Laboratório de Biomatemática, Faculdade de Medicina, Universidade de Lisboa, Avenida Professor Egas Moniz, Edifício Egas Moniz, ala C, Piso 0, 1649-028 Lisboa, Portugal; 5Instituto de Medicina Preventiva e Saúde Pública, Faculdade de Medicina, Universidade de Lisboa, Avenida Professor Egas Moniz, Edifício Egas Moniz, ala C, Piso 0, 1649-028 Lisboa, Portugal

**Keywords:** prevention, community-based intervention, food policy, healthy snacks, school

## Abstract

Policy interventions in the school food environment can improve dietary behaviors. However, the literature describing its development and implementation is scarce. This manuscript aims to describe the process of co-creation, implementation, monitoring, and evaluation of a Healthy Snacks Policy, in the scope of Sintra Grows Healthy intervention. Through a community-based participatory research methodology, the co-creation of the Healthy Snacks Policy comprises six stages: snacks evaluation, feedback sessions, class assemblies, school community assemblies, school cluster policy approval, and process evaluation. Within one school year, a Healthy Snacks Policy was co-created, approved, incorporated in the school regulations, implemented, continuously monitored, and evaluated. Regarding snacks evaluation, 1900 snacks were evaluated at the beginning of the school year and 1079 at the end of the school year. There were three feedback sessions, twenty-two class assemblies, and three school community assemblies. Most teachers perceived that children began to consume healthier snacks (72%); 66% of the children were considered to have started eating healthier; and most families said “yes or sometimes” when asked whether their children started requesting healthier snacks (70%), trying new foods (63%), and noticing improvements in their eating habits (74%). The co-creation of a Healthy Snacks Policy establishes an approach to effectively implement existing guidelines for school food supplies, complying with national priority implementation recommendations.

## 1. Introduction

The United Nations Convention on the Rights of the Child establishes children’s rights to protection, housing, education, health, healthcare, and adequate nutrition [1]. Also, the European Union Strategy on the Rights of the Child [2] reinforces the premise that nutrition during childhood is crucial for physical and cognitive development, well-being, health promotion, and disease prevention, influencing all subsequent life cycle stages [3].

According to the Global Burden of Disease study, in 2019, dietary risks ranked as the fifth risk factor that drives the most death and disability combined, among the Portuguese population [4]. Simultaneously, the other leading risk factors are also influenced by dietary habits, namely high fasting plasma glucose, high blood pressure, high body mass index, alcohol use, and high low-density lipoprotein (LDL) cholesterol levels [4]. Therefore, implementing effective public policies is of the utmost importance to promote healthy food environments [5], reducing the risk of chronic diseases, such as obesity, and decreasing dietary and health inequities [6].

In the past decade, there has been a growing emphasis on food and nutrition policies in Portugal. A key event to highlight includes the creation of the National Program for Healthy Eating Promotion (PNPAS) in 2012, aimed at improving the nutritional status of the Portuguese population, promoting the physical and economic availability of healthy foods, and creating conditions for the population to appreciate, value, and consume them on a daily basis [7]. Another key event occurred in 2017 when the Integrated Strategy for the Promotion of Healthy Eating (EIPAS) was published as a Law, putting into practice the principle of Health in All Policies [8]. From these two key events, improving food availability in schools was defined as crucial [7,8].

Children spend a significant part of their day at school, making it one of the ideal environments to positively influence their eating habits through curricula and policies that promote access to healthy foods [9]. Evidence suggests that food and nutrition policies are essential in transforming school environments into health-promoting settings [10].

In 2013, mandatory guidelines for school lunches were set in Portugal [11]. A guide for food supply in school buffets or cafeterias, as well as vending machines, was also created [12] and recently reinforced to establish mandatory rules for food availability in school buffets or cafeterias [13] and to assist schools in choosing the healthiest food alternatives. Typically, public primary schools in Portugal have canteens and do not have school buffets or cafeterias. Hence, children often eat school lunch, and generally bring mid-morning and afternoon snacks from home or receive them from the municipality. Snack consumption typically occurs during the school breaks either on playground or in the classroom.

There is evidence showing that primary school children consume snacks high in fat, sugar, and salt in this setting, contributing to inadequate dietary habits and excessive daily energy intake [14,15]. In 2021, a Portuguese guide for families, teachers, and educators, was created to raise awareness and promote healthy snacks among children and young people [16], categorizing foods as “foods to promote”, “foods to limit”, and “foods to avoid” in the school context. However, these guides remain non-mandatory and lack monitoring and evaluation of compliance. Thus, foods brought from home to be consumed at school are not monitored or regulated.

The municipality of Sintra, in alignment with the national guidelines, has been implementing several initiatives aimed at fostering a healthier school food environment, such as ensuring the quality of school lunches, with all menus devised by registered dietitians, implementing and monitoring a salt control policy, providing ongoing professional training for cooks, promoting family dinners at schools annually, and organizing healthy culinary workshops [17]. Also, Sintra’s municipality provides the option to purchase nutritionally balanced snacks [18] through placing a pre-order and paying a symbolic fee in accordance with the social support rank: EUR 0.34 when purchased with lunch, EUR 0.40 without lunch, EUR 0.17 for those in social support rank B, and free of charge for families in social support rank A.

Despite these efforts, there is a notable prevalence of overweight in school-aged children [19,20,21] and empirical knowledge of an excessive energy intake in morning snacks brought from home [22]. Consequently, a significant amount of plate waste at lunch, which is consistent with findings in similar contexts [23,24], may be depriving children of school lunch benefits and nutrition education opportunities. In recognition of the need for complementary strategies, Sintra’s municipality promotes Sintra Grows Healthy (SGH), along with partnerships between health and academic entities as stakeholders. Briefly, SGH consists of a multi-component and comprehensive school-based intervention for the promotion of healthy lifestyles, following a community-based participatory research methodology [17].

Considering the lack of rules on foods brought to school from home and the needs assessment performed, one of the intervention axes of SGH consists of the co-creation, implementation, monitoring, and evaluation of a Healthy Snacks Policy in the context of public primary schools. Evidence shows that school food policies, such as the provision of healthy foods and limiting unhealthy foods by restricting the availability of sugar-sweetened beverages, candies, and processed salty snacks in school, can significantly improve dietary behaviors, increasing fruit intake, and reducing the consumption of sugary drinks and processed snacks [25]. However, the literature describing the details of the process of development and implementation is scarce [26].

Thus, to fill in the gaps regarding the lack of rules for foods brought from home to be consumed at school and the dearth of narrative manuscripts that detail the process of development and implementation, this study aims to describe the co-creation, implementation, monitoring, and evaluation of a Healthy Snacks Policy, in the scope of the SGH intervention.

## 2. Materials and Methods

The SGH intervention began in the 2017/2018 school year, with one school cluster composed of three schools as the intervention group, and another school cluster as the control group with similar characteristics to the intervention group, selected at the schools’ convenience. SGH is now being implemented in eighteen school clusters and will soon extend to the entire population—20 school clusters (approximately 13,400 children)—with a total of 83 primary schools from the municipality of Sintra. The study protocol of SGH intervention has been previously detailed [17].

Focusing on the process of co-creation of the Healthy Snacks Policy, it comprises the following stages: snacks evaluation, feedback sessions, class assemblies, school community assemblies, school cluster policy approval, and process evaluation, which are hereby described. In the control group, only snack evaluation was carried out.

### 2.1. Snacks Evaluation

Quantitative evaluations of the nutritional quality of snacks consumed in the school context were conducted by trained researchers at the beginning (October and November) and at the end (June) of the school year, corresponding to pre- and post-intervention assessments. While the intervention was implemented only in the intervention group, the control group was included solely for comparison purposes, with no intervention applied. Snack consumption was directly observed during consumption in the classroom, in both groups, and data were recorded using specific registration grids. SGH’s methodological protocol standardizes food portions based on a published food weight and portion manual [27]. The nutritional composition of the snacks is calculated using a database developed by SGH, aligned with the reference values established in the Portuguese Food Composition Table [28]. The initial food list comprised 962 items from the Portuguese Food Composition Table and had been updated with several other food items (using the nutrition information presented on the product label), mainly comprising ready-to-eat items (such as cookies, cakes, and sugar-sweetened beverages), resulting in a list of 2479 food items. At the beginning of the school year, children’s snacks were assessed on two different days of the same week, while at the end of the school year, only one day was considered.

### 2.2. Feedback Sessions

The feedback sessions consist of meetings with the teachers, wherein the SGH technical team shares the results regarding the nutritional quality of the snacks, evaluated by class, school, and school cluster. Considering the findings, teachers are encouraged to reflect on and discuss the next steps to improve the nutritional quality of the snacks consumed in the classes they lead. In the context of community-based participatory research, these feedback sessions align with best practices for promoting high retention rates by fostering strong relationships between participants and the study team, ensuring that the participants perceive the benefits of the study, maintaining regular communication, and actively involving community-based and faith-based organizations [29].

### 2.3. Class Assemblies

Another intervention axis of SGH consists of a food literacy curriculum—Health at the table [30]—wherein the teacher leads weekly food education classes, typically lasting around 60 min. In the class assembly children and teachers get together to discuss the quality of the snacks they eat. The discussion is based on questions like “What is a healthy snack?” and “What reasons lead to the consumption of unhealthy snacks?” as well as “What could be implemented to improve snacks?”. Following the debate, the students write out the meeting minutes, which is then signed by the whole class, outlining the main points of discussion that will be later covered in an assembly with the entire school community.

### 2.4. School Community Assembly

The school community assemblies consist of meetings with families, teachers, and school staff, in which the SGH technical team shares not only the results regarding the nutritional quality of the snacks evaluated but also the class assemblies performed and their results, as well as a first draft of what could be the Healthy Snacks Policy for that school cluster, aligned with national guidelines. The whole school community is encouraged to contribute to and discuss the healthy snacks policy under construction, aiming to improve children’s eating habits. Afterwards, a deadline to revise and contribute to the document is established to achieve the final Healthy Snacks Policy for approval. Since not all families are present at the assembly, the outcomes of the school community assemblies are shared with all families, allowing everyone the opportunity to review and provide input.

### 2.5. School Cluster Policy Approval

The Healthy Snacks Policy is presented both at General and Pedagogical Councils to be approved by the various stakeholders of Sintra’s educational system and integrated into the school cluster’s regulations. The General and Pedagogical Councils are key school management bodies, with the General Council overseeing strategic decisions and community representation, while the Pedagogical Council coordinates and supervises pedagogical activities and educational guidance [31]. Once approved, the Healthy Snacks Policy is implemented, becoming part of the school regulations and identity, and is monitored daily by children and teachers and by the SGH technical team twice a year.

### 2.6. Process Evaluation

The SGH intervention entails process evaluation directed at every member of the school community to collect input from all parties involved in the pursuit of ongoing improvement. It includes questions for families, children, teachers, and school staff that are both closed and open-ended. Closed questions on a Likert scale cover the different intervention axes, namely the school food environment, whereas open-ended questions are meant to generate ideas and opportunities for improvement.

### 2.7. Ethical Issues

This study was conducted in accordance with the Declaration of Helsinki and approved by the Lisbon Academic Medical Centre ethics committee (401/17), the National Data Protection Commission (11468/2017), and the Ethics Boards of each participating school. Written informed consent was obtained from all subjects involved in the study.

## 3. Results

In the scope of the SGH intervention, within one school year, a Healthy Snacks Policy was co-created, approved, incorporated in the school regulations, implemented, continuously monitored, and evaluated. The process of co-creating the Healthy Snacks Policy is shown in Figure 1. The participants consisted of 510 children and 22 teachers from three primary schools in one school cluster during the 2017/2018 school year. Another school cluster, consisting of 488 children and 20 teachers from three primary schools, was the control group. While this manuscript focuses on a single school year involving one school cluster, it is noteworthy that seventeen additional school clusters have already replicated this intervention. Although the impact on the snacks’ nutritional quality is out of the scope of this manuscript, the quality of snacks consumed in the school setting has been improving and will be explored in a future publication.

Regarding snacks evaluation, during the 2017/2018 school year, 1900 snacks were evaluated at the beginning of the school year and 1079 at the end of the school year.

Feedback sessions, class assemblies, school community assemblies, school cluster policy approval, and process evaluation only occurred in the intervention group. The presentation of the snack evaluation findings was performed in three feedback sessions held in each school, twenty-two class assemblies, one for each class, and three school community assemblies, one for each school. Figure 2 provides an overview of the contributions made by children during class assemblies. The results from the class assemblies highlight that children identify a range of factors contributing to the consumption of unhealthy snacks, such as parental influence, lack of awareness, and convenience. They also propose several proactive measures, including creating snack rules, increasing fruit consumption, educating themselves and their parents about healthy choices, and actively involving themselves in the selection and preparation of their snacks.

Once the Healthy Snacks Policy was approved, it was written into an official document, incorporated into the school cluster’s regulations, and implemented. Considering the lack of rules for foods brought from home to be consumed at school, the Healthy Snacks Policy aims to regulate the context, quality, and quantity of the snacks children consume. The official document summarizes facts about children’s unhealthy eating habits and emphasizes the school–family commitment’s guiding principles (such as the need for schools to have environments that promote health and the role that families and schools play as collaborators in helping students adopt and maintain healthy lifestyles). Then, it explains the importance of snacks and what their adequate energy contribution should be, considering the school-aged children’s dietary energy requirements. Also, rules regarding the context of snack consumption are presented (e.g., it is recommended that snack consumption occurs in a class context, supervised by the responsible teacher, 10 min before playground time). Given the prevalence of children’s birthday celebrations during the school year and that most families bring food for the entire class, the Healthy Snacks Policy also specifies guidelines for birthday celebrations at school. It is recommended that birthday celebrations be held during the mid-afternoon snack period. Soft drinks are not allowed, and the birthday cake should be homemade and devoid of cream. Moreover, the Healthy Snacks Policy includes a food traffic light system that groups the foods into three categories: green (foods to promote consumption in the school context, such as whole grain bread, fresh fruit, vegetables, milk, yogurt, and cheese), yellow (foods to limit consumption in the school context to a maximum of once a week, namely, cereal bars, homemade cakes, and fruit nectars), and red (foods not to be consumed in the school context, i.e., soft drinks, pastries, chocolates, chips, and sausages). Finally, to monitor compliance with the Healthy Snacks Policy, children, supervised by their teachers, fill out a daily self-regulation snack evaluation chart based on the traffic light system. Additionally, guidance on filling out the monitoring chart for the Healthy Snacks Policy is displayed.

Regarding the process evaluation conducted, most teachers perceived that, after implementing the Healthy Snacks Policy, children began to consume healthier snacks (72%), and they reported that the Snack Evaluation Chart was an appropriate measure for children to help them to consume healthier snacks (82%). Concordantly, 66% of the children were considered to have started eating healthier. Concerning the school staff, 40% stated that the Healthy Snacks Policy led to children consuming healthier snacks (Figure 3). Most families said “yes or sometimes” when asked whether their children started requesting healthier snacks (70%) and trying new foods (63%) and whether they noticed improvements in their eating habits (74%) after the implementation of the Healthy Snacks Policy. Furthermore, 88% fully agreed or agreed that it is important to implement a Healthy Snacks Policy at school (Figure 4).

Figure 5 gathers information regarding families’ improvement suggestions for the Healthy Snacks Policy (*n* = 49). The improvement suggestions were gathered into ten categories: more diversity (*n* = 14), more quantity (*n* = 12), incentive to continue (*n* = 7), more flexibility (*n* = 6), less flexibility (*n* = 3), improve parents’ awareness (*n* = 2), share more recipes (*n* = 2), more appealing snacks (*n* = 1), more free foods (*n* = 1), and a higher snack quality (*n* = 1).

## 4. Discussion

This manuscript describes the process of co-creation, implementation, monitoring, and evaluation of a Healthy Snacks Policy in the context of public primary schools, filling in the gap regarding the lack of rules for foods brought from home to be consumed at school, and the dearth of narrative manuscripts on this topic. A detailed process description, as presented in this manuscript, ensures the intervention model’s replicability and transferability [25].

The evidence suggests that implementing primary school nutrition policies is linked to an increase in the availability of healthier foods or a decrease in the availability of unhealthy ones, improving the school food environment and, consequently, dietary behaviors [25,32]. Specifically, a 2018 systematic review and meta-analysis demonstrated that school food environment policies, such as the provision of healthy foods and competitive food standards, can significantly impact children’s dietary behaviors. For example, the direct provision of fruits and vegetables in schools increased children’s daily fruit intake by 0.27 servings and their combined fruit and vegetable intake by 0.28 servings. Additionally, food and beverage standards, which limit the availability of unhealthy snacks, reduced the consumption of sugar-sweetened beverages by 0.18 servings per day and unhealthy snacks by 0.17 servings per day [25]. Another systematic review by Grigsby-Duffy et al. (2020) [32] emphasized that school nutrition policies can positively transform the school food environment by increasing the availability of healthy options and reducing unhealthy foods. Given the growing global burden of diseases linked to poor dietary habits and the unique opportunity that schools have to influence children’s eating behaviors, these policies represent a feasible and promising strategy for improving diet quality [32]. Implementing standards or regulations to ensure the nutritional quality of snacks could positively impact children’s dietary choices. Nevertheless, the implementation of regulations concerning school-provided snacks and snacks brought from home varies significantly across countries, often lacking systematic monitoring [33]. In addition, robust evidence shows that community engagement interventions improve several health outcomes across multiple conditions, even though the most successful community engagement model is still unknown [34]. The involvement of children, teachers, school staff, and families in the policy development process is crucial to enhance the long-term sustainability of the intervention, fostering a sense of ownership. Furthermore, through the co-creation process and the implementation of the Healthy Snacks Policy, new rules are established and incorporated into the regulations, becoming a part of the school’s identity.

The community-based participatory research methodology followed is a highly valuable research approach that focuses on enhancing health equity [35] and improving health by increasing research quality, empowerment, capacity building, and sustainability [36]. SGH class and community assemblies, as well as the feedback sessions, are crucial parts of this methodology. Additionally, the feedback loops established through class and community assemblies allowed for continuous policy refinement, ensuring its relevance and effectiveness.

Rigorous impact evaluation is needed to establish a sustainable intervention model. Therefore, the SGH intervention includes data collection, requiring parents’ consent [17]. Even though not all parents provide permission for data collection, the Healthy Snacks Policy is applied to all children within the school cluster, as it is the school’s responsibility to ensure a healthy food environment.

Moreover, to monitor compliance with the Healthy Snacks Policy, children, supervised by their teachers, fill out the snack evaluation chart daily using the traffic light system. This procedure promotes children’s self-regulation, which is described as a predictor of better achievement, interpersonal behaviors, mental health, and healthy living in adulthood [37]. Through this intervention, the children becomes an active agent of change, as evidenced by their participation in class assemblies, where they engage in discussions about the quality of their snacks, propose solutions for improvement, and take responsibility for communicating these ideas to the broader school community and establishing commitments to improve their snacks. Also, the children, who learned about the snack traffic light system and monitor their daily snacks, persuade family members at home to provide healthier snacks. Simultaneously, since research indicates that social influence plays a crucial role in shaping children’s decision-making abilities, and that children generally adhere to group decisions, it is important to consider that children can influence and be influenced by their peers [38].

In addition, the Healthy Snacks Policy, which included a list of foods to promote, foods to limit, and foods not to consume, is particularly important as an intervention that can establish specific rules regarding foods to be consumed and foods that should not be made available and that can improve the nutritional quality of snacks consumed by children in the school context [33].

Regarding 2017/2018 process evaluation, most teachers and families believed that, after the implementation of the Healthy Snacks Policy, children began to consume healthier snacks; most children also held the perception that they started eating healthier. Although the intervention does not directly target the school staff and, with the implementation of the Healthy Snacks Policy, the snack consumption happens in classrooms with teachers rather than on playgrounds with school staff, 40% of the school staff still recognized an improvement in children’s snack consumption. 

Concerning the family’s suggestions for improvements in the Healthy Snacks Policy, the most frequent suggestions were for snacks to be more diverse, have greater quantity, and for the initiative to continue. In the scope of the following public procurement for school meals, a more diverse menu was established and implemented for the snacks provided by the municipality. It includes new foods such as nuts, baby carrots, cherry tomatoes, peanut butter, rusks, rice cakes, corn cakes, and a variety of breads with cheese, ham, butter, or jam, alongside milk, yogurt, or 100% fruit juice. Also, the SGH intervention is still ongoing; it started with one school cluster as the intervention group in the 2017/2018 school year, as described, and is currently being implemented in eighteen school clusters (2024/2025). In the next school year, 2025/2026, we hope to have reached all of Sintra’s primary school clusters (*n* = 20), totaling more than thirteen thousand children. Additionally, paradoxically, some families suggested more flexibility while others asked for less flexibility in school snack rules. Despite the challenges of satisfying everyone, the SGH team will keep working on the best solutions, combining scientific data with practical experience. Another family’s improvement suggestions were to improve parents’ awareness and share recipes. Therefore, the SGH technical team developed an interactive e-book “Healthy and Sustainable Snacks: A Practical Guide” [39], which covers topics related to childhood food and nutrition and nutritionally balanced snacks, provides tips on how to choose foods and prepare healthy snacks, and includes a menu covering two weeks of morning and afternoon snacks, recipes, and a quiz.

Ensuring health promotion is a responsibility of the municipalities [40], which are in a unique position to influence health outcomes through “top-down” policy interventions and “bottom-up” interactions with local stakeholders, due to their connections with a particular population and geographic area [41].

The synergy between Sintra’s municipality and the academic partners is crucial and strategic, as it promotes evidence-based practices and allows rigorous monitoring and evaluation to take place. This partnership enables the timely and accurate return of data to the school communities in real time, which is not always carried out while conducting integrated fieldwork [26], as well as objective measurements by trained researchers following established protocols [17]. Simultaneously, it allows the production and dissemination of scientific evidence based on practices. The recruitment of a multidisciplinary team by the municipality, including registered dietitians and physical activity, health, and education professionals, is also a determining factor, facilitating communication and cooperation with local actors and stakeholders.

Nevertheless, with all efforts and investments, childhood obesity remains a global public health challenge, with growing trends [42]. Poor-quality school food, limited physical activity options, and low health literacy, are recognized determinants of early-life origins of childhood obesity [43]. The SGH intervention aimed to tackle these issues and is aligned with early interventions to address obesity, promoting healthier lifestyles and school food environments [43].

The process of co-creation, implementation, monitoring, and evaluation of a Healthy Snacks Policy in the context of public primary schools faces several challenges and potential obstacles. One of the main potential challenges is the involvement of teachers, who often feel overwhelmed, particularly in primary education, where they teach under a single-teacher model, face a demanding formal school curriculum, and are frequently asked to participate in multiple projects that are often not integrated with one another. The SGH intervention is not intended to be an isolated initiative, but rather an integrated approach designed to support and complement the formal school curriculum. Also, some teachers argue that children’s nutrition should be solely the responsibility of families, considering it outside their professional duties. In this context, we emphasize the significance of health-promoting schools, and, through the implementation of the co-created policy, we seek to strengthen the teacher’s autonomy and legitimacy in enforcing nutrition-related rules within the school setting, just as they would with any other rules outlined in the school regulation. Another limitation is the low engagement and participation of families. To address the fact that families who would benefit the most—typically those from low-income groups—tend not to participate, the assembly method is used in conjunction with a strategy where teachers send the information shared during assemblies to all families through a more tailored channel [44].

Based on our experience, for the successful adaptation of this intervention methodology across different contexts, it is essential to have a dedicated technical team to lead the intervention and ensure effective project management. This team should work in close collaboration with local school teams, municipalities, and parent associations. According to the Healthy Food Environment Policy Index (Food-Epi) from Portugal, in 2022, one of the priority implementation recommendations is to ensure the effective implementation of existing guidelines for school food supply by defining a model for monitoring compliance with existing standards or guidelines [45]. Within one school year, a Healthy Snacks Policy with rules and a strategy for monitoring and evaluating snacks consumed in the school context was co-created, approved, incorporated in the school regulation, implemented, continuously monitored, and evaluated, establishing an approach to effectively implement the existing guidelines for school food supply, and complying with national priority implementation recommendations. Nevertheless, this policy is not an isolated initiative, as it is part of a multi-component intervention (SGH) that combines policies, environmental rules, and health literacy educational strategies into the curriculum. By addressing dietary behaviors in a structured, community-engaged framework, SGH sets a precedent for developing health-promoting school environments that can be replicated, contributing to a growing body of evidence supporting early interventions in school settings to tackle childhood obesity. Future research should focus on longitudinal studies to track the long-term impact of these interventions for broader implementation.

## Figures and Tables

**Figure 1 nutrients-16-03374-f001:**
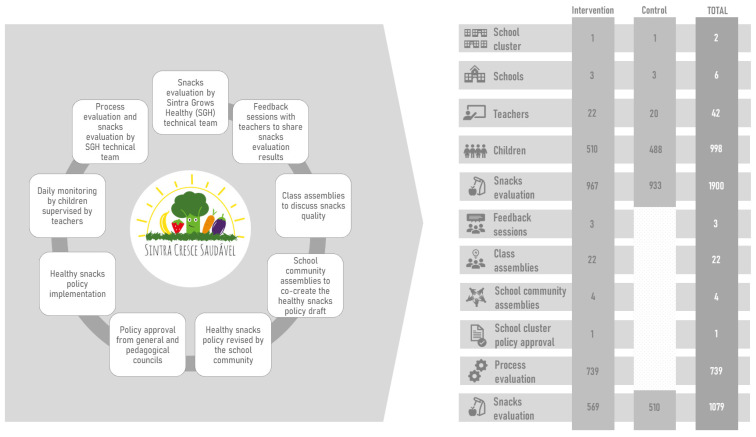
Co-creation of the Healthy Snacks Policy.

**Figure 2 nutrients-16-03374-f002:**
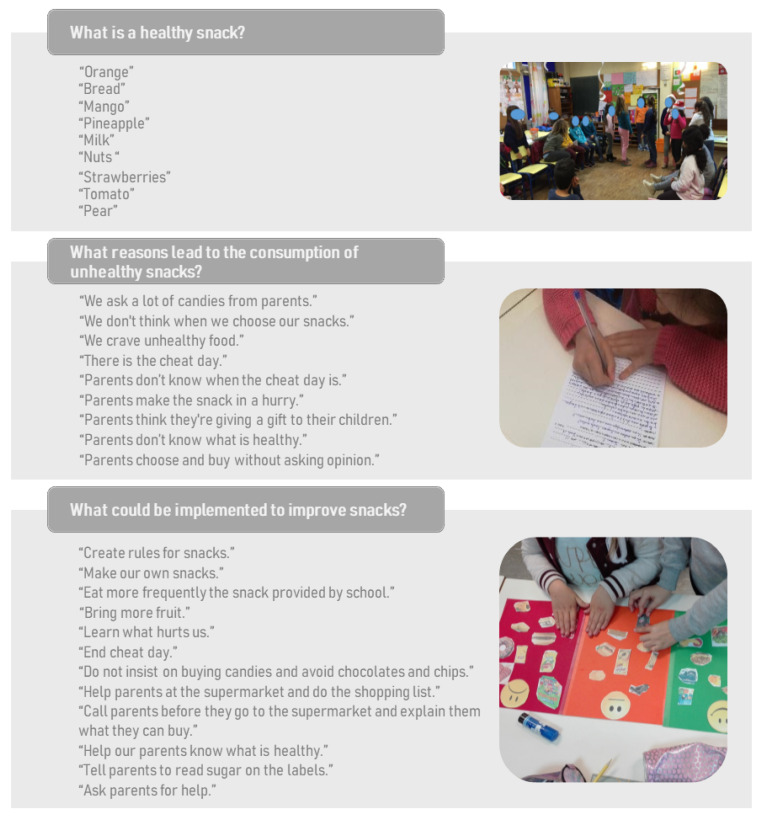
Results from class assemblies.

**Figure 3 nutrients-16-03374-f003:**
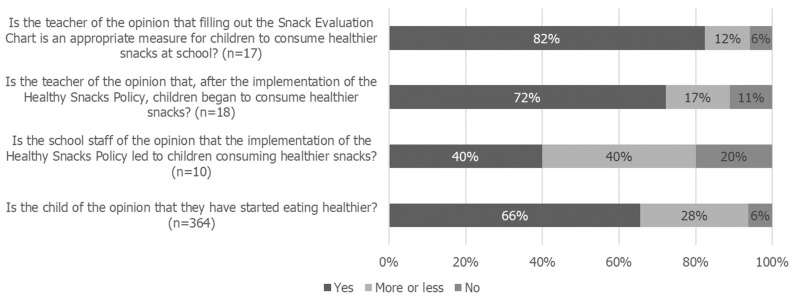
Process evaluation for teachers, school staff, and children.

**Figure 4 nutrients-16-03374-f004:**
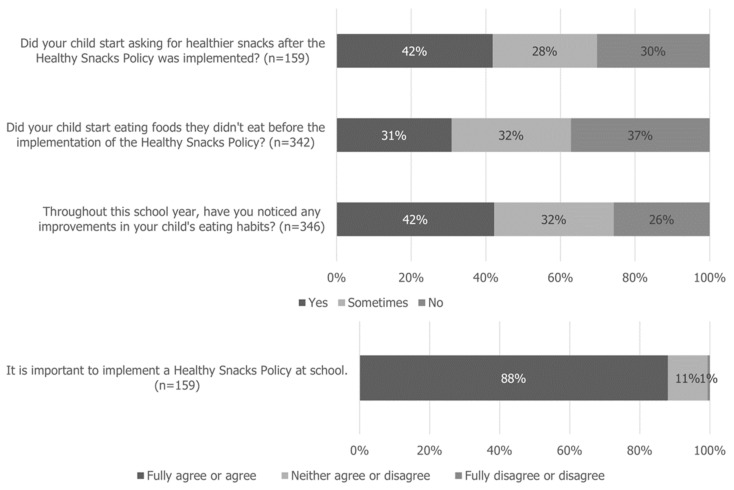
Process evaluation for families.

**Figure 5 nutrients-16-03374-f005:**
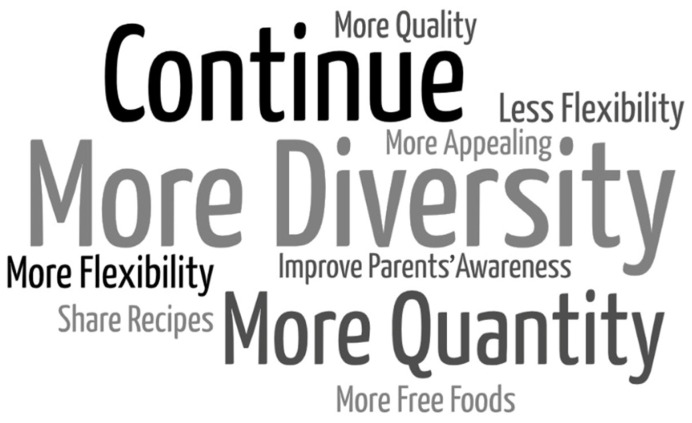
Families’ improvement suggestions for the Healthy Snack Policy.

## Data Availability

The data presented in this study are available on request from the corresponding author due to ethical reasons.

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
