# Peer review of "Co-Creation and Implementation of a Healthy Snacks Policy in Primary Schools: Data from Sintra Grows Healthy"

_nutrients, 2024, doi:10.3390/nu16193374_

Round 1

Reviewer 1 Report (Previous Reviewer 4)

Comments and Suggestions for Authors

The authors adequately addressed previous concerns of the reviewer.

Author Response

Reviewer 2 Report (Previous Reviewer 2)

Comments and Suggestions for Authors

I acknowledge that you have adressed all the issues of reviewers.

Author Response

Reviewer 3 Report (Previous Reviewer 1)

Comments and Suggestions for Authors

Authors have answered reviewer questions appropriately and the manuscript is improved.

Author Response

Reviewer 4 Report (New Reviewer)

Comments and Suggestions for Authors

Dear authors, thank you for submitting the manuscript entitled ‘Co-creation and implementation of a Healthy Snacks Policy in Primary Schools: data from Sintra Grows Healthy’ The research question is important in the field of applied research on healthy eating in childhood and interesting and relevant data are presented here. However, there are some issues that need to be addressed in more detail.

There are parts of the manuscript in a different font colour suggesting that the latest version of the manuscript was not submitted - this should be rechecked.

L103 The results of the policy intervention should be described in more detail in the introduction or discussion section, where this is more appropriate.

L106-109 Was the aim also to obtain results on the actual consumption of (healthy) snacks in terms of quality and quantity? Or define as non-objective if not addressed/published in this context.

Add information on the importance of pre-post evaluation of school snack (consumption?). How was the impact measured - i.e. what is the endpoint/measure of effectiveness of the intervention?

The discussion section should also include more concrete results from other studies/policies. E.g. L269 the extent to which dietary policies are able to increase/decrease the availability of healthy and unhealthy foods.

Was familiarity with food/snacks also asked about? e.g. do some children not know certain healthy foods that could be used as snacks - more knowledge could increase intake of such foods e.g. Mere Exposure Effect considered?

More detail in the description of the method is necessary and relevant. Provide concrete numbers on how many numbers were reached in each group.

Was ethical approval obtained (children and parents?) - add this information to the methods section.

L135 Why was only one day considered at the end of the intervention? Explain this in the methods section or discussion.

L136 Add information about how intensively the teachers were able to perceive the consumption of snacks and for how many children at a time? Were the teachers in the classes during the snack break?

L148 How long did these nutrition education lessons last, e.g. 10 minutes, 30 minutes, a whole lesson or more? This is also important for other interested schools/organisations/countries to know. 

L155 Include specific figures on how many families, teachers and school staff were involved in the study. 

Include specific snack categories that were evaluated for the 1900 snacks. How is a snack defined - add information here.

L313 the food lists: which foods are beneficial and which are not should be included in the article (at least part of it)

L357 Reference in brackets (38) vs [38]

Author Response

This manuscript is a resubmission of an earlier submission. The following is a list of the peer review reports and author responses from that submission.

Round 1

Reviewer 1 Report

Comments and Suggestions for Authors

This manuscript describes an implementation for a policy procedure change within schools for home-provided snack foods. While this manuscript is well-organized, there are some minor suggestions that would improve the manuscript. Overall the discussion overextends some of the results, specifically suggesting that children are agents of change within this intervention. While this may be true, there are no data presented to suggest that and this point could be indicated as a future direction. There are also no limitations discussed.

Introduction

-          Line 86, what is meant by a symbolic fee?

-          What do you mean by empirical knowledge of excessive energy intake in morning snacks brough from home (line 89). Does this mean the families lack this knowledge, or have this knowledge?

Methods

-          Why were two days evaluated in school beginning, but only 1 day at end?

Comments on the Quality of English Language

Overall there are some grammar issues with tenses, with the methods section being written in present, rather than past tense.

Reviewer 2 Report

Comments and Suggestions for Authors

I am particularly impressed by your approach to a participatory process and its innovative application in the context of a school environmental initiative. Nevertheless, further insight into the potential challenges and obstacles that may arise from such an approach would have been beneficial. Additionally, an evaluation of the anticipated outcomes, given that the study commenced in 2017/18, would have enhanced the readers' understanding. Furthermore, recommendations on how this approach could be adapted for diverse contexts would have been valuable. 

Reviewer 3 Report

Comments and Suggestions for Authors

I regret to say that this study should not be published in its current state, as it has notable shortcomings in all its sections that make it inadvisable. The study does not describe an adequate scientific process and the results do not have sufficient guarantees. The following are the most notable shortcomings:

- The study does not present a minimally acceptable review of the state of the art, only the general context of the study. There is only one reference to similar studies (25), which is a review with meta-analysis of 91 previous studies from 2018, including 129 references. The results of this study alone indicate to the authors how deep and broad the literature in this area is, of which nothing is presented and it is also stated that ‘literature describing the details of the process of development and implementation is scarce’, which is not true. The authors should carry out such a thorough review of the state of the art, since they are ignoring practically all the scientific advances of the last few years in their field of study.

- On a topic as prolific as EU healthy eating policies, one would expect a much more recent literature than the one presented, with hardly any scientific references after 2020 and no citation of the latest developments in the field. The authors should take the best and most recent references to substantiate their study.

- The objectives of the study are not clearly described, so it is not clear to the reader what is being assessed and how the study is measured.

- There is no section to describe the population and sample, so the reader does not know who is being evaluated to have a clear reference of what is being done in the study.

- 2.1. Snacks evaluation is a poorly described heading. Reference is made to a previous study, but it is not possible to know what was done and what was concluded in that initial phase without consulting the named reference (17). The authors should describe in detail what was done in that preliminary phase. Why and why it was done, what was discovered and what was concluded as a result, as well as describing the process followed.

- 2.2. Feedback sessions is another poorly described heading. Nothing is described about these feedback sessions. Without this information provided by the authors, it is impossible to assess the scientific process.

- 2.3. and 2.4., as there is no section devoted to describing the population and sample, it is not clear what is intended by these actions and whether they are appropriate for the target population.

- 2.5. It is stated in this section that the study was approved by a higher committee, the code for this approval and the name of the committee should be clearly stated (apart from the final section on ethical approval).

- 2.6. Nothing is known about the reliability and validity of the assessment instruments used, nor about their selection process.

- In the Results, the sample is mentioned for the first time and a control group is mentioned, of which it is not known what it consists of or what scientific safeguards have been included, and an evaluation of snacks which is not part of this study. Hardly has this research been described and there is already talk of parts of it being left for other publications. This is a bad decision for this publication.

- The results consist of collecting feedback from participants, which is an indirect measure of what has been achieved. It is a very poor and indirect measure of the achievements of the research. Moreover, it is unknown whether this opinion is adulterated by the benefits and time investment in the intervention proposal, as any control measures of the study are unknown.

- There is no statistical analysis of the results. Neither correlational, nor inferential, not even descriptive tables of the results or visual references apart from a few simple Figures. This is not a valid way of collecting scientific results.

- The discussion shows mostly studies that were not in the previous theoretical framework, which is an unwanted disconnection of theoretical framework and discussion that should be avoided.

- There are no sections dedicated to the limitations of the study or to the conclusions.

Reviewer 4 Report

Comments and Suggestions for Authors

In the Materials and Methods section, 1) make sure to be clear whether the SGH intervention is still on going. If it is not, the following sections (i.e., 2.1-2.6) need to be changed to the past tense; 2) please add 1 brief sentence describing difference between treatment and control--in the results, it's clear that the control did not receive the interactive components but this is not clear in Methods.

Comments on the Quality of English Language

Minor grammatical errors need to be corrected: Line 53: include should be "includes"; line 57: should change "in a daily basis" to "on a daily basis"; line 70: change from "children often eat the school lunch" to "children often eat school lunch" Line 86: change from "purchase nutrition balanced" to "purchase nutritionally balanced"; Line 114 and 116: change "snacks evaluation" to "snack evaluation"; Line164: change "of SGH intervention" to "of the SGH intervention"

Round 2

Reviewer 3 Report

Comments and Suggestions for Authors

none of the key points of the study have been sufficiently improved. Among other things (not the only ones) it can be said:

- No scientifically validated instruments are used.

- There is no adequate literature review.

- The variables involved in the study are not defined and exploited.

- There is no inferential statistical analysis to contribute scientific knowledge to the field of study.

- The recommendations suggested in the Discussion and other sections have not been deeply followed.
